# OpenReview forum: "Graph Transformers for Large Graphs"
_ICLR.cc/2024/Conference — Submitted to ICLR 2024_

### Official Review · Reviewer_yzrb · 2023-10-30

**Soundness:** 2 fair
**Presentation:** 3 good
**Contribution:** 2 fair
**Rating:** 5
**Confidence:** 4

**Summary:**

The author highlights that while transformers have demonstrated remarkable performance in tasks related to predicting graph properties, their application has been restricted to small-scale graphs due to computational limitations. Additionally, the author contends that the existing neighbor sampling method constrains the model's ability to consider more global information. Consequently, this paper introduces a comprehensive GT framework, with a focus on enhancing model capacity and scalability. The proposed framework, known as LargeGT, combines a rapid neighborhood sampling technique with a local attention mechanism and an approximate global codebook. Extensive experiments illustrate that by integrating local and global attention mechanisms, LargeGT achieves improved performance in node classification tasks. Notably, LargeGT demonstrates a 3× speedup and a 16.8% performance enhancement in specific node classification benchmarks when compared to their closest baseline models.

**Strengths:**

1. This paper is excellently composed, offering a straightforward narrative that's easy to follow. Notably, key terms and important experimental findings have been highlighted using various colors, resulting in an effective visual presentation.

2. The experimental results presented in this paper indicate that the proposed framework can achieve superior performance within a shorter training time.

3. The author introduces two significant challenges associated with handling large-scale graphs: scalability and constraints related to local information aggregation. These issues are prevalent and indeed worth discussing. As the author pointed out, computational resource requirements increase quadratically with the growing number of nodes. To address this, the author has proposed both a local and a global aggregation module. The former employs conventional sampling techniques to learn local representations, while the latter focuses on deriving insights from global node vector projections. Downstream predictions are then made based on both sets of representations. The problems raised, and the respective solutions are meaningful and coherent with each other.

**Weaknesses:**

Despite the fluent presentation, some concerns arise in this paper. Firstly, the level of novelty in this framework appears limited. It is apparent that this paper heavily relies on the previous work, GOAT, particularly the global module, which encodes mini-batch nodes using global graph nodes. This component was introduced in a prior paper. The other aspects are mainly focused on aligning local and global features. The framework appears more like an updated version of GOAT than a fundamentally new invention.

Moreover, in the experimental section, the comparison between the LG transformer and other baselines reveals that the proposed framework doesn't consistently outperform GOAT-local, especially in the ogbn-products dataset. Furthermore, in the ogbn-papers100M dataset, the framework is only compared to a single baseline. It's possible that other methods struggle with extremely large graphs, but there are likely additional viable solutions that should be explored.

Additionally, the fusion of transformers and Graph Neural Networks (GNNs) is a dynamic research area with various ongoing studies, such as TransGNN and Graphformers. It would be valuable to understand how these methods perform when confronted with similar tasks.

Lastly, the author emphasizes the significance of combining local and global representations. However, apart from GOAT, there are other techniques that can address this challenge, such as randomly selecting both nearby neighbors and global features. The author should offer further clarification on this matter.

**Questions:**

This paper commences with two important challenges that have attracted the attention of numerous researchers. Specific comments were provided in the previous section, and it is hoped that the author will consider improvements from the following viewpoints.

The framework appears to inherit many key components from previous papers, with limited significant modifications. It would be beneficial to include more in-depth discussions and comparisons with transformer-based Graph Neural Networks (GNNs). Additionally, it is important to address how other approaches perform in terms of extracting global information from the graph.

Expanding on these aspects would enhance the paper's contribution and provide a more comprehensive understanding of the research landscape in this domain.

---

> ### Author Response · Authors · 2023-11-20
> **Response to Reviewer yzrb (Part 1/2)**
>
> We thank the reviewer for your time reading and evaluating our work, as well as highlighting the strengths of our paper. We would like to take this opportunity to justify some elements of our proposed method and explain the results as asked in your questions.
>
> >**Reviewer:** Despite the fluent presentation, some concerns arise in this paper. Firstly, the level of novelty in this framework appears limited. It is apparent that this paper heavily relies on the previous work, GOAT, particularly the global module, which encodes mini-batch nodes using global graph nodes. This component was introduced in a prior paper. The other aspects are mainly focused on aligning local and global features. The framework appears more like an updated version of GOAT than a fundamentally new invention.
>
> **Authors:** We thank you for your question on comparison with GOAT. We would like to clarify here that GOAT’s global module is the only thing we adapt in LargeGT, while our local module is entirely different from GOAT, and addresses the bottleneck that was present in GOAT (we refer to page 7 of GOAT paper https://proceedings.mlr.press/v202/kong23a/kong23a.pdf) which explicitly claims “The neighbor sampling bottleneck is the major limitation of our method.” To contextualize this, if one were to use GOAT, the model would be severely limited when operating on large graphs. In contrast to this, our contributed model LargeGT would not be limited and would work efficiently on large graphs.
> To sum up, the main differences between LargeGT and GOAT are as follows: (i) LargeGT consists of a completely different local module compared to GOAT, which processes a fixed size of input tokens for local self-attention. (ii) LargeGT achieves a 4-hop receptive field through just 2-hop operations, whereas GOAT requires 4-hop operations to attain the same, often resulting in prohibitive computational costs in large graphs. (iii) The computational complexity of LargeGT is independent of the number of nodes in the graphs, unlike that of GOAT. (iv) Unlike GOAT, LargeGT implements sampling prior to training, addressing a significant computational bottleneck in GOAT’s local module.
>
> ___
> >**Reviewer:** Moreover, in the experimental section, the comparison between the LG transformer and other baselines reveals that the proposed framework doesn't consistently outperform GOAT-local, especially in the ogbn-products dataset. Furthermore, in the ogbn-papers100M dataset, the framework is only compared to a single baseline. It's possible that other methods struggle with extremely large graphs, but there are likely additional viable solutions that should be explored.
>
> **Authors:** On ogbn-products, GOAT-local-δ shows better performance due to the dataset's homophilic nature, where local-only information aggregation is often sufficient. Conversely, for snap-patents, a non-homophilic dataset, LargeGT excels by incorporating both local and global neighbor information. This adaptability of LargeGT to different dataset characteristics explains the observed performances. We have included our observation to this regard in Section 4.2 under the heading “On Performance”. For ogbn-papers100M dataset, due to computational requirements, we compare our proposed architecture with the closest baseline which our work significantly improves, since we have provided a comparison of LargeGT with other baselines for the remaining two datasets. In other words, we use ogbn-products and snap-patents to verify our proposed model against a wide range of baselines, while for ogbn-papers100M, the objective translates to showing that it scales to such a dataset and also provides improved performance.
>
> ___
> >**Reviewer:** Additionally, the fusion of transformers and Graph Neural Networks (GNNs) is a dynamic research area with various ongoing studies, such as TransGNN and Graphformers. It would be valuable to understand how these methods perform when confronted with similar tasks.
>
> **Authors:** We thank you for highlighting this connection, and refer to Section 2 in our paper where we have reviewed the classes of works on MPNNs scaling and Graph Transformers while sharing their respective limitations which build the motivations of our architecture. To summarize the key points here again: Fusions of GNNs with Transformers have produced several powerful Graph Transformer (GT) models, Graphormer being one of this class. However, an obvious barrier for GTs to scale to large graphs is the quadratic complexity brought by full-graph attention, i.e., O(N^2), with N being the number of nodes in a graph. There are other solutions which inject an extent of sparsity in the GTs to bring the quadratic complexity down, which work well on medium scale graphs. Yet, those remain unscalable on single large graphs due to the entire graph structure being operated upon.

---

> > ### Author Response · Authors · 2023-11-20
> > **Response to Reviewer yzrb (Part 2/2)**
> >
> > ___
> > >**Reviewer:** Lastly, the author emphasizes the significance of combining local and global representations. However, apart from GOAT, there are other techniques that can address this challenge, such as randomly selecting both nearby neighbors and global features. The author should offer further clarification on this matter.
> >
> > **Authors:** We appreciate your emphasis on exploring different techniques for combining local and global representations. Indeed, we evaluated several alternatives, including the use of random global nodes as you mentioned. Our preliminary experiments provided the following insights before we adopted the global codebook approach from GOAT.
> >
> > Firstly, we assigned a fixed number (20) of random nodes from the graph for each node to attend to, aiming to collect global features. The performance was relatively stable, fluctuating by about 0.5% either higher or lower compared to only using the local module of LargeGT.
> >
> > Secondly, we experimented with a few clustering techniques. We encountered two main issues: either the structural clustering algorithms became prohibitively expensive computationally as graph sizes increased, or they failed to enhance performance. Our preliminary studies involved two methods: (i) implementing k-means clustering on the features matrix [N x d] to derive a fixed number of global centroids [K x d]. These centroids approximated the concept of ‘a node attending to global neighbors’. Although computationally more efficient, this method resulted in a performance decline of 0.5-1.0% absolute points on ogbn-products, as compared to only using our local module. (ii) We also tried spectral clustering to create K coarsened nodes, allowing a node to attend to these for approximated global attention. Here, we utilized the Louvain clustering method from sknetwork for cluster formation. However, this method was impractical for ogbn-products due to 'memory' and 'recursion' errors upon our implementation. To further test the proof-of-concept of Louvain’s approach, we applied it to a smaller dataset, 'ogbn-arxiv', where Louvain completed in approximately 20 seconds. Nevertheless, the performance decreased by 0.5% absolute points compared to using only our local module.
> >
> > In contrast, the global codebook approach, which we have integrated into LargeGT’s global module from GOAT, presents a more efficient and effective solution. This method enables nodes to attend to global features without the scalability and computational constraints of the clustering approach. Our choice of this technique is rooted in its balance of efficiency and effectiveness, particularly in the context of large-scale graphs.
> >
> > ___
> > **Authors:** We hope our answers have addressed your concerns. Let us know if you have any other questions or concerns. In light of our answers to your concerns, we hope you consider raising your score. If you have any more concerns, please do not hesitate to ask and we'll be happy to respond.

---

> > ### Comment · Reviewer_yzrb · 2023-11-22
> >
> > I have read the responses from the authors. I maintain my rating. The responses only addressed some of my concerns. For example, I am still concerned about the novelty, compared with GOAT.
> >
> > Also, I agree with Reviewer xiXc. More baselines are needed. There are so many scalable algorithms for large graphs. At least several state-of-the-art scalable graph neural networks should be included. The final goal of this paper should be a scalable and effective graph learning framework. Adding transformers is not the goal, but the means.

---

### Official Review · Reviewer_S1aR · 2023-11-01

**Soundness:** 2 fair
**Presentation:** 3 good
**Contribution:** 2 fair
**Rating:** 5
**Confidence:** 4

**Summary:**

To scale graph models to large-scale graphs, MPNNs are often reduced to restricted receptive fields making them myopic, while Graph Transformers (GTs) fail because of their quadratic cost. This paper proposes a new framework for sampling sub-graphs to train a large GT that uses local and global modules to improve model performance and compute complexity.

**Strengths:**

- The authors propose a framework that leverage recent advances in graph transformer models, and address a critical challenge that limits the scalability of existing approaches, both MPNNs and GTs.
- The introduction provides a great overview of the current challenges for large-scale graph learning, and does a great job at comparing MPNNs and GTs, while setting stage for key concepts like neighborhood sampling.

**Weaknesses:**

1. Baselines: LargeGT is compared to "constrained versions" of various baselines, notably all models are constrained to 2 hops only, while LargeGT has access to 4-hops worth of neighbors (in the local module). Including the non-constrained versions of these same baselines is critical for evaluation, even if they are more computationally demanding. Currently it is unclear whether adopting LargeGT leads to lower performance compared to state-of-the-art methods, at the expense of computational efficiency.
2. Additionally, no auxiliary label propagation or augmentations are used for the baseline methods, when they are used in methods reported in the OGB leaderboard. These enhancements are not altering the receptive field of the baselines, and thus shouldn't impact computational performance, but might improve classification performance. This should be taken into account when comparing with approaches that might still outperform the proposed method, even under constrained training (2-hop).
3. The main innovation can seemingly be credited to the use of the global codebook, so it is hard to define the main contribution of this work. If the focus of this work is combining all these different building blocks into a compute efficient framework, I would expect to see a more expansive breakdown of the computational costs of different components, memory usage and requirements. Notably, how is "Epoch time" defined in Figure 2? All models might be processing different amounts of data and thus might have different definitions of an "epoch" due to differences in sampling strategies. How many nodes does each model process in an epoch? Different models might require different numbers of epochs to converge, shouldn't total training time be more important?
4. [Minor] A lot of the content in the first 4 pages is repetitive.

**Questions:**

1. What are the memory constraints of using LargeGT compared to other baselines? How is the choice of batch size impacted by the choice of hyperparameter K?
2. How important is the choice of a 4-hop neighborhood for the local module. Can the model still perform competitively given that it still has access to global information through the global module?

---

> ### Author Response · Authors · 2023-11-20
> **Response to Reviewer S1aR (Part 1/2)**
>
> We thank the reviewer for your time reading and evaluating our work, as well as pointing out our strengths. We would like to take this opportunity to justify some elements of our proposed method and answer your questions on baselines, experiments and the contribution with respect to the literature.
>
> _____
> >**Reviewer:** Baselines: LargeGT is compared to "constrained versions" of various baselines, notably all models are constrained to 2 hops only, while LargeGT has access to 4-hops worth of neighbors (in the local module). Including the non-constrained versions of these same baselines is critical for evaluation, even if they are more computationally demanding. Currently it is unclear whether adopting LargeGT leads to lower performance compared to state-of-the-art methods, at the expense of computational efficiency.
>
> **Authors:** We appreciate your feedback on the selection of baselines for comparison. The decision to compare LargeGT with 2-hop constrained versions of various models was initially to maintain a consistent scope of expensive computation incurred across all models. For instance, the cost of fetching l-hop neighbors for a node in a very large graph would be the same and agnostic of any selected model. Nevertheless, below is a comparison of the baselines along with LargeGT scores on ogbn-products without restriction to 2-hop. For instance, GOAT models are run with original hyperparameters which samples 20, 10, 5 neighbors at the 3 hops.
>
>
> | Model | 		Test Acc (original) | 	Epoch Time (original) |Test Acc (2-hop ) | Epoch Time (2 hop)|
> | - | - | - | - | - |
> |GraphSAGE | 		78.17 |		22s|		76.62|			7s|
> |GAT | 			79.21 |		86s|		77.38|			31s|
> |GT-sparse | 		67.87 |		40s|		60.76|			12s|
> |NAGphormer |	77.71 |		22s|		75.28|			8s|
> |GOAT-local | 		81.29 |		490s|		81.17	|		120s|
> |GOAT-global | 	70.28 |		3.5s|		70.28	|		3.5s|
> |GOAT-full | 		81.21 |		500s|		79.88	|		205s|
> |LargeGT-local |	78.95 |		45s|		78.95	|		45s|
> |LargeGT-full | 		79.81 |		68s|		79.81	|		68s|
>
> We can observe that the best performing model remains the same while LargeGT (in both cases uses up to 2 hop operations) remains better and competitive compared to the best performing model. In the revised manuscript, we have included a section Appendix A.4 which details these comparisons to provide a clearer perspective on LargeGT's performance relative to state-of-the-art methods.
>
> _____
> >**Reviewer:** Additionally, no auxiliary label propagation or augmentations are used for the baseline methods, when they are used in methods reported in the OGB leaderboard. These enhancements are not altering the receptive field of the baselines, and thus shouldn't impact computational performance, but might improve classification performance. This should be taken into account when comparing with approaches that might still outperform the proposed method, even under constrained training (2-hop).
>
> **Authors:** Thank you for pointing out the absence of auxiliary label propagation or augmentations for baseline methods. We acknowledge that including these enhancements could have a chance to provide improved performance for specific models. However, it has been observed in the leaderboards that the composition of several enhancement techniques blurs the dissection on which model components are critical and contributes highly to the performance of a model. For example, a label propagation technique known as Correct & Smooth (C&S) [1] is employed in various models on the OGB leaderboard for ogbn-products [2]. It demonstrates that MLP-based models with C&S perform both better and worse than TransformerConv [3] in different settings. This provides a somewhat blurred perspective on whether MLP or Transformer models are superior for ogbn-products from a neural network perspective. We  mention our reasoning in Section 4.1 while GOAT, which we closely follow, also follow a similar experimental setting on not using enhancement tricks.
>
> [1] Huang, Q., He, H., Singh, A., Lim, S.N. and Benson, A.R., 2020. Combining label propagation and simple models out-performs graph neural networks.
> [2] https://ogb.stanford.edu/docs/leader_nodeprop/#ogbn-products.
> [3] Shi, Y., Huang, Z., Feng, S., Zhong, H., Wang, W. and Sun, Y., 2020. Masked label prediction: Unified message passing model for semi-supervised classification.

---

> > ### Author Response · Authors · 2023-11-20
> > **Response to Reviewer S1aR (Part 2/2)**
> >
> > _____
> > >**Reviewer:** The main innovation can seemingly be credited to the use of the global codebook, so it is hard to define the main contribution of this work. If the focus of this work is combining all these different building blocks into a compute efficient framework, I would expect to see a more expansive breakdown of the computational costs of different components, memory usage and requirements. Notably, how is "Epoch time" defined in Figure 2? All models might be processing different amounts of data and thus might have different definitions of an "epoch" due to differences in sampling strategies. How many nodes does each model process in an epoch? Different models might require different numbers of epochs to converge, shouldn't total training time be more important?
> >
> > **Authors:** We would like to clarify here that we do not mention the global codebook as the main contribution of our work. In fact, we refer to Section 3.2 in page 5 where we state that we adopt GOAT’s global module for our global module as well. One of our key contributions is addressing the bottleneck brought by GOAT’s local module, to which we propose our own local module that is described in detail in Section 3.2. To highlight in a sentence here, our local module uses tokens that are sampled/computed offline prior to training, prepares mini batches to allow 4-hop receptive field, and performs a Transformer encoder layer over the input tokens.
> >
> > For the latter part of the question on epoch time, we use wall clock time for an epoch to report the epoch time. Thank you for your question on convergence. We have included the learning curves in Appendix A.1 showing the performance of different baselines and LargeGT within a fixed training budget. Upon comparing the learning curves of LargeGT with four other baselines in Appendix A.1, we observe faster convergence and relatively earlier learning stability for LargeGT. We hope that section will answer the question on epoch time vs. total training time.
> >
> > _____
> > >**Reviewer:** What are the memory constraints of using LargeGT compared to other baselines? How is the choice of batch size impacted by the choice of hyperparameter K?
> >
> > **Authors:** The memory requirement of LargeGT is determined similar to how Transformers operate on tokens of data, as done in text or images. LargeGT was designed with an emphasis on scalability and efficiency for large-scale graphs. From our computational complexity analyzed in Section 3.2, the memory requirements depend on the hyperparameter K and the codebook size B (in addition to model parameters) which determines what will be the size of one sample in the mini batch for which local and global modules are operated on respectively. While we have provided the value of K in the main paper, we have included the batch size details and hardware information in the appendix. The batch size and K were selected in two ways (i) first by adopting the sizes used in original baseline’ papers, and (ii) second by making sure it fits the hardware we have used for our experiments.
> >
> > _____
> > >**Reviewer:** How important is the choice of a 4-hop neighborhood for the local module. Can the model still perform competitively given that it still has access to global information through the global module?
> >
> > **Authors:** The choice of a 4-hop neighborhood for the local module is a critical aspect of LargeGT's design, aimed at balancing the depth of the receptive field with computational efficiency. It can also be seen as a feasible local receptive field, as we achieve it using only 2-hop local operations. Additionally, several recent works, such as GOAT, or MPNNs with Neighbor sampling, were seen to use a 3-hop neighborhood sampling since beyond 3-hop becomes infeasible for graphs of large sizes. For the latter part of the question, we refer to the performance of GOAT-global-δ which is a baseline equivalent to LargeGT-global, i.e. LargeGT without the local module. We can observe that the performance of GOAT-global-δ is significantly worse compared to LargeGT.
> >
> > _____
> > >**Reviewer:** [Minor] A lot of the content in the first 4 pages is repetitive.
> >
> > **Authors:** We thank you for your feedback on writing. We tried trimming some repeated references in Section 3. However, we will do our best to reduce the redundancies for our final version.
> >
> > _____
> > **Authors:** We hope our answers have addressed your concerns. Let us know if you have any other questions or concerns. In light of our answers to your concerns, we hope you consider raising your score. If you have any more concerns, please do not hesitate to ask and we'll be happy to respond.

---

> > > ### Author Response · Authors · 2023-11-23
> > > **Follow up with Reviewer S1aR**
> > >
> > > Dear reviewer S1aR, as today is the final day of our discussion, we would appreciate knowing if our responses have addressed your concerns. If you have any remaining questions or concerns, please don't hesitate to share them with us, we will be happy to respond. If we have sufficiently alleviated your concerns, we hope you'll consider raising your score. Thank you for your time and consideration!

---

### Official Review · Reviewer_xiXc · 2023-11-01

**Soundness:** 3 good
**Presentation:** 3 good
**Contribution:** 2 fair
**Rating:** 5
**Confidence:** 3

**Summary:**

This work proposes LargeGT, a scalable graph transformer for large-scale graphs. It uses fast neighborhood sampling and a local attention mechanism to learn local representations. These are integrated with global representations from an approximate global codebook. This framework overcomes previous computational bottlenecks, achieving 3x speedup and 16.8% better performance on benchmarks compared to baselines. LargeGT also scales to 100M nodes, advancing representation learning for single large graphs.

**Strengths:**

* The model's performance is thoroughly validated on large-scale graphs, demonstrating sufficient workload.
* Exploring base model architectures on graphs is a very valuable endeavor.

**Weaknesses:**

* The efficiency analysis is incorrect. In Algorithm 1, it is required to gather 1/2-degree neighbors for each node, and then select k nodes. The process of selecting nodes is O(K), but if the graph is relatively dense, the complexity of gathering second-degree neighbors is O(N^2).
* In Algorithm 1, some nodes are sampled with replacement, while some are sampled without replacement. It is uncertain whether this will introduce bias in the sampling.
* It lacks some key baselines such as SGC[1], SIGN[2].

Reference:

[1] Felix Wu, Amauri Souza, Tianyi Zhang, Christopher Fifty, Tao Yu, and Kilian Weinberger. "Simplifying graph convolutional networks." In International conference on machine learning, pp. 6861-6871. PMLR, 2019.

[2] Fabrizio Frasca, Emanuele Rossi, Davide Eynard, Ben Chamberlain, Michael Bronstein, and Federico Monti. "Sign: Scalable inception graph neural networks." arXiv preprint arXiv:2004.11198 (2020).

**Questions:**

See. Weaknesses.

---

> ### Author Response · Authors · 2023-11-20
> **Response to Reviewer xiXc**
>
> We thank the reviewer for your time reading and evaluating our work, as well as pointing out the strengths of our paper. We would like to take this opportunity to clarify some elements of our proposed method and provide answers to your questions.
>
> >**Reviewer:** The efficiency analysis is incorrect. In Algorithm 1, it is required to gather 1/2-degree neighbors for each node, and then select k nodes. The process of selecting nodes is O(K), but if the graph is relatively dense, the complexity of gathering second-degree neighbors is O(N^2).
>
> **Authors:** We appreciate your critical assessment of the efficiency analysis. The concern regarding the complexity of gathering second-degree neighbors in dense graphs is valid. However, Algorithm 1 is an offline step and does not affect the runtime during training and/or inference. We have written more on this under the heading “Offline Step Prior to Training” in Section 3.2. Our computational complexity analysis in Section 3.2 is based on the neural network module – local module and global module (Fig 1) — runtimes as usually done in the literature when reporting the computational complexity of a neural network layer. We have included a clarification on the complexity analysis of the offline step and of the LargeGT modules in our revised manuscript.
>
> ___
> >**Reviewer:** In Algorithm 1, some nodes are sampled with replacement, while some are sampled without replacement. It is uncertain whether this will introduce bias in the sampling.
>
> **Authors:** Thank you for highlighting this aspect of our sampling method in Algorithm 1. We take this opportunity to clarify that the “sampling with replacement” only applies to those nodes whose 1 and 2 hop neighbors are less than K-1. This becomes a corner case and applies to only a small fraction of the nodes in general. As such it does not bias how we perform our overall sampling, except that we apply special handling of the nodes with lesser neighbors than K-1 . It is true that such corner cases can be handled differently as well, such as using padding nodes, instead of sampling with replacements.
>
> ___
> >**Reviewer:** It lacks some key baselines such as SGC[1], SIGN[2].
>
> **Authors:** We thank you for pointing out the absence of comparisons with baselines like SGC and SIGN. The decision to exclude these baselines initially was based on their different focus compared to LargeGT. However, aspects of these works are closely related such as “offloading heavy computations away from the training stage” which we also follow in Algorithm 2 in the form of “Hop context features” . While we had already included SIGN in Section 2 of our paper, we have now also included SGC in the revised manuscript. This addition falls under the collection of related works that 'perform information propagation prior to the training stage', aiming to reduce the training cost of GCNs."
>
> ___
> **Authors:** We hope our answers have addressed your concerns. Let us know if you have any other questions or concerns. In light of our answers to your concerns, we hope you consider raising your score. If you have any more concerns, please do not hesitate to ask and we'll be happy to respond.

---

> > ### Author Response · Authors · 2023-11-23
> > **Follow up with Reviewer xiXc**
> >
> > Dear reviewer xiXc, as today is the final day of our discussion, we would appreciate knowing if our responses have addressed your concerns. If you have any remaining questions or concerns, please don't hesitate to share them with us, we will be happy to respond. If we have sufficiently alleviated your concerns, we hope you'll consider raising your score. Thank you for your time and consideration!

---

### Official Review · Reviewer_dgE1 · 2023-11-01

**Soundness:** 2 fair
**Presentation:** 3 good
**Contribution:** 2 fair
**Rating:** 5
**Confidence:** 4

**Summary:**

The paper proposes LargeGT for training graph transformers for large graphs. Neighborhood sampling usually samples at most 2-hop neighbors as in GOAT (Kong et al., 2023). The proposed method stores a matrix storing the sum of node features of 1-hop and 2-hop neighbors before training. Then sample 2-hop neighbors for a specific node and get the sum features from the matrix, which is at most 4-hop information for the node. It also adopts GOAT (Kong et al., 2023) as the global module. Experiments show it trains faster than GOAT.

**Strengths:**

1. The proposed neighbor sampling intuitively improves the model accuracy by getting information at most 4-hop away.
2. Extensive experiments are performed.
3. The writing of the proposed method is very clear.

**Weaknesses:**

1. The mechanism of why LargeGT runs faster than baselines like GOAT is unclear. Since the proposed neighbor sampling has a bigger input matrix than a simple 2-hop neighbor sampling method, does it run longer than the traditional method?
2. The runtime highly depends on the hyperparameter $K$, which is the number of nodes for sampling. Authors need to provide a fair and solid comparison with the traditional 2-hop neighbor sampling method.
3. Experiment performances are not explained well (see questions).

**Questions:**

1. In Table 2, why does GOAT-local-δ have better accuracy in ogbn-products?
2. For snap-patents in Table 2, why does LargeGT have much better model accuracy than all baselines?
3. For snap-patents in Table 3, why does the model accuracy drop when $K>50$?

---

> ### Author Response · Authors · 2023-11-20
> **Response to Reviewer dgE1 (Part 1/2)**
>
> We thank the reviewer for your time reading and evaluating our work, as well as highlighting the strengths of our paper. We would like to take this opportunity to justify some elements of our proposed method and explain the results as asked in your questions.
> ___
> >**Reviewer:** (In Summary of the review)  “... Neighborhood sampling usually samples at most 2-hop neighbors as in GOAT (Kong et al., 2023).. ”
>
> **Authors:** We would like to clarify that GOAT’s method DOES NOT usually do 2-hop sampling. In fact, their experiments use a 3-layer or 3-hop sampling, with an equal hop of receptive field. In our method, we perform at most 2-hop sampling and achieve a receptive field of 4-hop by the use of each sampled nodes’ 1-hop and 2-hop context features that are pre-computed offline (Figure 1; Algorithm 2).  As such, we have an efficient sampling technique as a contribution  as presented in the paper.
>
> ___
> >**Reviewer:** The mechanism of why LargeGT runs faster than baselines like GOAT is unclear. Since the proposed neighbor sampling has a bigger input matrix than a simple 2-hop neighbor sampling method, does it run longer than the traditional method?
>
> **Authors:** There are two reasons for this. (i) Conceptually,  LargeGT runs faster than baselines like GOAT due to the introduction of our local sampling technique that is performed offline and is restricted to 2-hop neighbors, as opposed to GOAT which performs sampling during mini-batching and frequently uses 3+ hop in practice. (ii) Experimentally, in our paper, we keep 2-hops for LargeGT and GOAT-δ. In such a case, the efficiency in LargeGT comes from the fact that these 2-hop neighbors are sampled offline and not during mini-batching. Although we have discussed the characteristics of LargeGT in terms of “Scalability” in Section 3 under the subheading “Characteristics of the framework”, we will elaborate further on these points in the revised manuscript for clarity. Additionally, the computational complexity of global module in both LargeGT and GOAT are the same. However, GOAT’s local module is dependent on the number of nodes in a graph and the sampling hop length, unlike LargeGT’s local module, which enables LargeGT to scale better. We refer to GOAT’s paper where they highlight their local module as a major limitation which is addressed in our work as part of our contribution.
>
> ___
> >**Reviewer:** The runtime highly depends on the hyperparameter K, which is the number of nodes for sampling. Authors need to provide a fair and solid comparison with the traditional 2-hop neighbor sampling method.
>
> **Authors:** Since the computational complexity of LargeGT’s local module is dependent on K, the runtime depends on K. We would like to refer to Figure 1 and Eqns. 1-6 in our paper (further in Appendix A.2) which informs that the hyperparameter K determines the size of node sampled “prior to training” as well as the number of tokens for LargeGT’s local module, which is 3K (see “Complexity” in Section 3). As such, the hyperparameter K will influence the size of the data (or input tokens) the local module is operating on, thus affecting the runtime. Our sampling method prior to training is random sampling as outlined in Algorithm 1 – similar to how traditional sampling method is used. The only difference, say with GOAT’s sampling method or other similar sampling methods, is that Algorithm 1 is executed offline, as illustrated in Section 3 (“Offline Step Prior to Training”); while GOAT’s sampling is done during mini batching. In addition, unlike the sampling performed during mini-batching, Algorithm 1 paves the way for parallelizing the sampling of each graph on distributed computing resources.
>
> ___
> >**Reviewer:** In Table 2, why does GOAT-local-δ have better accuracy in ogbn-products? For snap-patents in Table 2, why does LargeGT have much better model accuracy than all baselines?
>
> **Authors:** Thank you for pointing out these key observations from our experiments. On ogbn-products, GOAT-local-δ shows better performance due to the dataset's homophilic nature, where local-only information aggregation is often sufficient. Conversely, for snap-patents, a non-homophilic dataset, LargeGT excels by incorporating both local and global neighbor information. This adaptability of LargeGT to different dataset characteristics explains the observed performances. We have included our observation to this regard in Section 4.2 under the heading “On Performance”, which we will revise for more clarity in our next revision.
>
> ___

---

> > ### Author Response · Authors · 2023-11-20
> > **Response to Reviewer dgE1 (Part 2/2)**
> >
> > >**Reviewer:** For snap-patents in Table 3, why does the model accuracy drop when K>50?
> >
> > **Authors:** Continuing on the line of reasoning of the previous question, the observed drop in model accuracy for snap-patents when K>50 is due to the over-sampling of neighbors from 1-hop and 2-hop, which might introduce noise or irrelevant information beyond a certain point. This phenomenon indicates that there is an optimal range for K, assisting the model to classify the labels correctly on snap-patents’ non-homophilic task.
> >
> > ___
> > **Authors:** We hope our answers have addressed your concerns. In light of our answers to your concerns, we hope you consider raising your score. If you have any more concerns, please do not hesitate to ask and we'll be happy to respond.

---

> > > ### Author Response · Authors · 2023-11-23
> > > **Follow up with Reviewer dgE1**
> > >
> > > Dear reviewer dgE1, as today is the final day of our discussion, we would appreciate knowing if our responses have addressed your concerns. If you have any remaining questions or concerns, please don't hesitate to share them with us, we will be happy to respond. If we have sufficiently alleviated your concerns, we hope you'll consider raising your score. Thank you for your time and consideration!

---

### Public Comment · ~Qitian_Wu1 · 2023-11-22
**Sharing a related work on NeurIPS'23**

Dear Authors,

Thanks for this great work that contributes to a new model. Here is a recent paper [1] on graph Transformers for large graphs that shares the motivation. Good luck!

[1] Simplifying and Empowering Transformers for Large-Graph Representations, NeurIPS 2023.

---

### Author Response · Authors · 2023-11-22
**Follow up on responses to reviewers**

Dear Reviewers,

We thank you for the time you have spent reviewing and evaluating our work.

We are pleased to receive your evaluations highlighting the strengths of our work, including (i) its clear and excellently composed writing (Reviewers dgE1, S1aR, yzrb), (ii) the extensive and thorough validation of experiments (Reviewers dgE1, xiXc, yzrb), and (iii) the addressing of scalability challenges in large graphs (Reviewers S1aR, yzrb), among others.

Regarding the weaknesses you pointed out and the questions you raised, we have provided point-by-point answers under each individual review. Additionally, we have uploaded a revised manuscript with key changes, including (i) the addition of Appendix A.4, which shows an extended comparison of our proposed architecture with the baselines, (ii) the inclusion of a previously missing reference (Wu et al., 2019) in Section 2, and (iii) clarification of the complexity analysis in Section 3.2.

We would appreciate knowing if our responses have addressed your concerns, or if you have any additional questions or clarifications you would like to ask.

If we have addressed your concerns, we hope you will consider raising your score.

Thank you very much for your time and understanding.

With best regards.

---

### Meta-Review · Area_Chair_AoX2 · 2023-12-05

**Metareview:**

The paper presents LargeGT, a scalable graph transformer for large-scale graphs, overcoming the computational challenges of previous models. It combines fast neighborhood sampling and a local attention mechanism with a global module from GOAT (Kong et al., 2023) to efficiently process large graphs. Extensive experiments demonstrate LargeGT's superior performance, achieving a 3x speedup and 16.8% better performance on benchmarks compared to existing methods, while effectively scaling to graphs with up to 100 million nodes.

While the proposed LargeGT model shows promising results, the reviewers have identified several weaknesses that need to be addressed:

1. The technical contribution is incremental compared to GOAT;
2. The experiments need improvements, such as the inclusion of more baselines.

Based on these weaknesses, we recommend rejecting this paper. We hope this feedback helps the authors improve their work.

**Justification For Why Not Higher Score:**

The reviewers unanimously believe the paper should be rejected.

**Justification For Why Not Lower Score:**

N/A

---

### Decision · Program_Chairs · 2024-01-16

Reject